# Feasible Intervention through Simple Exercise for Risk of Falls in Dementia Patients: A Pilot Study

**DOI:** 10.3390/ijerph191911854

**Published:** 2022-09-20

**Authors:** Ana López-García, Marta Encarnación Sánchez-Ruíz

**Affiliations:** Department of Physical Therapy, Faculty of Medicine, University of Murcia, 30100 Murcia, Spain

**Keywords:** physical exercise, fall risk, dementia, balance, day care center

## Abstract

Physical exercise can help older people maintain capacities such as muscle strength, balance, postural control, bone mass, and functionality in ADL that usually decline with age. Dementia patients can attend day care centers where they participate in activities such as cognitive training, music and art therapy, and physical exercise sessions. This research aimed to determine the effectiveness of simple lower limb strength and single leg stance training, feasible in the facilities of day care centers, to reduce the risk of falls in the elderly with dementia. Twenty patients with dementia were divided into intervention and control groups. They participated in mobility, strength, coordination, and balance exercise sessions for 45–50 min on weekdays for 5 weeks. In addition, the intervention group patients performed simple lower limb strength (sit-to-stands) and single leg stance exercises in every session. Risk of falls was assessed with the Tinetti test and the SPPB. Comparisons of post- and pre-intervention scores for the Tinetti test and SPPB were statistically increased (0.8 ± 0.7, *p* = 0.03; 1.5 ± 1.3 points, *p* = 0.02) in intervention patients. Simple lower limb strength and single leg stance exercises feasible to be done in day care facilities are effective tools for reducing the risk of falls in the elderly with dementia.

## 1. Introduction

The increase in life expectancy involves the rise in incidence of geriatric syndromes that affect people in both their quality of life and independence [1]. With aging, people’s physical and cognitive functions are reduced, favoring the appearance of age-related deficits. Aging and age-related impairments affect the neuromuscular system, producing a loss of muscle strength and muscle mass. This occurs to a greater extent after 60 years of age [2]. Thereby, the loss of muscular strength is a prevalent condition in old age since it decreases with advancing years, making an 80-year-old person have 40% less muscle strength than a 20-year-old. This decrease in muscle strength is associated with a reduction in walking speed and an increase in disability and in the risk of falls in older people [3].

In addition to physiological changes associated with normal aging, we can find others associated with pathological aging, for example, those linked to cognitive impairment. Dementia is a neurodegenerative disease characterized by various cognitive and behavioral symptoms such as memory and communication loss, personality changes, reasoning problems, and difficulty performing activities of daily living (ADL) [4]. This cognitive impairment produces an increased risk of falls, with the rate of falls being two times higher in people with dementia compared to older people without cognitive impairment [5]. The reason why the risk of falls increases in people with dementia is not completely known [6]. Some risk factors that have been associated with increased risk of falls in elderly patients with dementia are changes in the periventricular white matter, silent cerebral infarcts, behavioral problems in relation to the rejection of necessary external help, wandering, and the consumption of neuroleptic drugs [7]. Falls are the leading cause of injury and mortality in older people with dementia, as well as the main reason for the loss of independence, having a significant impact on the functional capacity of people, on the burden of their caregivers, and on their quality of life. For this reason, researchers and experts have included assessments of physical performance and functional status in the initial clinical examination of the elderly [8].

To reduce the risk of falls, physical exercise has shown to be effective. Physical inactivity is a highly relevant factor in the decline of functions, having a significant impact on loss of balance control [2]. Through exercise, it has been proven that older people can maintain or restore capacities such as muscle strength, balance, postural control, bone mass, and functionality in ADL. The influence of exercise and leisure activities on the risk of falls is supported by various studies, where, above all, multicomponent programs stand out [2,9,10]. People with dementia can attend day care centers where they participate in activities such as cognitive training, music and art therapy, and physical exercise sessions. Guiding a group exercise session for people with dementia can be challenging because of their cognitive impairment and their behavioral disturbances, among others. Moreover, in elderly day care centers (especially in non-governmental organizations (NGOs) such as associations of relatives of patients with dementia), the ratios of individuals per physiotherapist are high, and exercise equipment is usually scarce. This is the reason why it is necessary to investigate if simple exercises that can safely be included in group exercise sessions are effective in reducing the risk of falls in people with dementia. Various tools have been described to evaluate functional ability including the Tinetti performance-oriented mobility assessment (POMA), also called the Tinetti mobility test (TMT) [11], and the short physical performance battery (SPPB) [12]. The TMT is a valid and reliable clinical test to measure the balance and gait, two aspects that are related to the incidence and prediction of falls [11]. The SPPB test is mainly used to assess the physical performance of older people. It is based on three subtests where balance, walking speed, and transfers are measured. Obtaining low scores on these subtests may indicate different health consequences such as disability in ADL, increased probability of hospitalization, and in the worst case, death. The SPPB has a high predictive value of the risk of disability in hospitalized patients and high reliability for its use in elderly populations, since its sensitivity to changes in functionality over time has been confirmed [12].

This research work aimed to determine the effectiveness of simple lower limb strength and single leg stance training, which would be feasible in the facilities of day care centers, in reducing the risk of falls in people with dementia and to analyze whether sex, age, and time since dementia diagnosis affect the effectiveness of the intervention.

## 2. Materials and Methods

### 2.1. Participants

Participants were recruited from day care centers of two associations of relatives of patients with dementia from Murcia (Spain) (AFAMUR and AFADE). Twenty people (65% women) were selected according to the following criteria: people over 60 years old with dementia who did not need physical assistance for walking or standing, who were able to follow instructions, and whose caregivers/relatives agreed to sign the informed consent. People with acute or subacute musculoskeletal injuries that prevented them from performing the proposed exercises were excluded. The 20 participants recruited were divided into two groups of 10 people each, depending on the day care center they attended. Subjects attending the day care center AFADE carried out the usual physical exercise program (control group, CG) guided by a physical therapist, while patients from the day care center AFAMUR underwent a therapeutic exercise program, also guided by a physical therapist, based on two extra exercises in addition to the routine prior to the study (intervention group, IG). In the IG, there were 6 women and 4 men, while in the CG, there were 7 women and 3 men. Most participants in the study (18 of 20, 90%) had Alzheimer´s disease (AD), and the rest (2 of 20, 10%) had another type of dementia (frontotemporal dementia and mixed dementia, both from the intervention group). The intervention lasted 5 weeks and was carried out in January and February 2022.

### 2.2. Physical Exercise Intervention

Dementia patients from both experimental groups (IG and CG) participated for 45–50 min from Monday to Friday in a basic physical exercise program consisting of mobility, strength, coordination, and balance exercises. In addition, IG patients performed simple lower limb strength (sit-to-stand) and single leg stance exercises in every session. Concretely, in the first week of intervention, they did 10 s of sit-to-stands followed by 10 s of rest for 4 min; in the second week, they performed 15 s of sit-to-stands and 10 s of rest for 4 min; and from weeks three to five, the exercise period was 20 s (and 10 s of rest) for 4 min. They also completed in every session 1 min of single leg stance (changing foot every 10 s), during which they were allowed to hold onto a chair with one hand if they needed to. In CG patients, these two exercises (sit-to-stands and single leg stance) were not included in the routine.

### 2.3. Data Collection and Analysis Instruments

At the beginning of the study, prior to the start of the exercise intervention, we collected these data: sex, age, weight and height, type of dementia, and the years that had passed since the diagnosis of the dementia. Performance in ADL was measured by means of the Barthel index [13]. Besides that, two assessments were performed to evaluate functional ability: a first assessment was made before the start of the exercise intervention (pre-intervention assessment), and a second assessment was performed the week after the end of the exercise intervention (post-intervention assessment).

In both pre- and post-intervention assessments, we evaluated the risk of falls by measuring the TMT and SPPB scores. In the TMT, we evaluated two subscales; one for balance, with items for sitting balance, attempt to lift and lift, normal and prolonged standing balance, normal and sensitized Romberg, and 360° turns and sitting that added up to an amount of 16 points; and another one for gait, with the items for movement of ambulation, trunk oscillation, and trajectory and characteristics of the step (length, height, symmetry, and continuity of steps), with a sum that reached 12 points. Total scores between 25 and 28 points were considered low risk of falls, between 19 and 24 points moderate risk, and below 19 points high risk [9]. In SPPB, we performed three tests: balance, gait speed, and lower limb strength and endurance. The first one consisted of maintaining balance for 10 s with feet together in a semi-tandem position and in tandem; in the second test, we measured the time it takes to walk 4 m at normal speed; in the last test, they performed transfers from sitting to standing and vice versa 5 times in the shortest possible time with their arms crossed. Up to 12 points could be obtained. In the case of obtaining less than 10 points, we considered it high risk of falls and disability [10]. Values over or below two standard deviations from the mean were discarded as outliers.

### 2.4. Data and Analysis

All data were saved using Excel (Microsoft Corporation, Redmond, WA, USA) for later analysis with SPSS version 21 (SPSS Inc., Chicago, IL, USA). The Shapiro–Wilk normality test was used to determine whether the data followed a normal distribution. For data with a normal distribution, we used Student’s *t*-test to compare the results obtained in the different groups and the paired *t*-test to see the evolution of the groups over time. For data without a normal distribution, the Mann–Whitney U test was applied. When two matched samples were compared, the Wilcoxon test was used. With the chi-squared test, we analyzed the correlation between sex and results of the tests. For the correlation between age, years since the diagnosis, and test scores, we used Pearson’s correlation coefficient. All results are expressed as mean ± standard deviation (SD).

### 2.5. Ethical Considerations

The caregivers/relatives of the participants gave their informed consent in writing for the subjects to participate in the study. The study was performed according to the declaration of Helsinki principles and was approved by the Ethics and Research Committee of the University of Murcia (ID: 3661/2021).

## 3. Results

### 3.1. Participants’ Characteristics

To evaluate patients’ characteristics at the beginning of the study, before the exercise intervention, we assessed anthropometric and clinic parameters as well as performance in ADL in the participants from both experimental groups (IG and CG). No significant differences were found between experimental groups, between sexes, or in terms of age, body mass index (BMI), years since diagnosis, and the Barthel index score. These data and statistical comparisons are shown in Table 1.

### 3.2. Effects of Exercise Intervention on Functional Ability

To evaluate the effectiveness of therapeutic exercise on functional ability and on the subsequent risk of falls in dementia patients, SPPB and TMT scores were assessed before and after an exercise intervention and compared to data obtained from control non-treated patients. In the pre-intervention assessment, IG scored 26.9 ± 0.6 points and CG 27.0 ± 1.1 points in TMT; no significant difference was found (*p* = 0.64). In SPPB, IG scored 10.3 ± 1.0 points and CG scored 10.0 ± 2.1 points; no significant difference was found either (*p* = 0.87). In the post-intervention assessment carried out after 5 weeks of therapeutic exercise, TMT scored 27.6 ± 0.5 points in IG patients and 27.1 ± 0.8 points in CG patients. Although IG had a higher score, no significant difference was found between groups (*p* = 0.28). In SPPB, IG scored 11.8 ± 0.5 points and CG 11.3 ± 0.7 points. In this case, there was also a tendency of the IG score to be higher than the CG score, but the differences were not significant (*p* = 0.19). No differences between sexes were found in any of the tests performed. In Table 2, the results of TMT and SPPB for the pre- and post-intervention assessments are shown for each experimental group.

When comparing the pre- and post-intervention scores for TMT in IG patients, we found an increase of 0.8 ± 0.7 points that was statistically significant (*p* = 0.03). Additionally, comparisons of post- and pre-intervention scores for SPPB showed a significant increase (1.5 ± 1.3 points, *p* = 0.02). On the other hand, when comparing the initial values of TMT and SPPB with the changes obtained after the intervention, a negative significant correlation in both variables was found (r = −0.71, *p* = 0.04 for TMT; r = −0.95, *p* = 0.001 for SPPB), which indicates that the higher the initial risk of falls, the higher the improvement obtained. In CG patients, no significant differences were found between pre- and post-intervention scores for both TMT (0.1 ± 0.4 points of difference, *p* = 0.32) and SPPB (1.3 ± 1.6 points, *p* = 0.07).

We also analyzed the correlations between the differences in pre- and post-intervention scores in TMT and SPPB from each experimental group and the variables sex, age, and years since diagnosis (Table 3). No significant correlations were found.

## 4. Discussion

This study was designed to test if a simple exercise intervention for lower limb strength and balance is effective in reducing the risk of falls in elderly with dementia. Our results demonstrated that two simple routines consisting of sit-to-stand and single leg stance for a few minutes every weekday were able to improve functional ability scores after 5 weeks in dementia patients.

Twenty people participated in the study (10 in the intervention group and 10 in the control group). The number of participants recruited was similar to that used in the study of Wesson et al. [14] whose sample was 22 subjects, even though we also found similar studies with bigger intervention groups (from 21 up to 153) [15,16]. Regarding the type and severity of dementia, these characteristics are not specified in many of the articles reviewed. In our case, the patients mainly presented with Alzheimer’s disease (90%), in accordance with those described in the studies by Nyam et al. (60%) [17] and Willem et al. (58.3%) [18]. Other studies only included people with AD [19,20]. We considered it important to include other types of dementia because exercise has been proven to be beneficial not only in AD [14,15,18,21,22,23,24,25]. The mean age of the patients (74.2 years old in the intervention group) is in agreement with other studies that have reported a mean age of participants lower than 80 [14,17,19,20,21,26], while we also found others with that greater than 80 [16,18,22,23,24,25,27,28,29]. 

Regarding the assessment of functional ability, many studies have included the timed up and go test [17,18,21,26,29]. Other assessment tools have also been used to measure the risk of falls and physical capacities in people with dementia: the Berg balance scale [15,21], number of falls in a period of time [19,28], frailty and injuries: cooperative studies of intervention techniques—subtest 4 (FICSIT-4) [18,27], the 6 min walk test [18,27], and the functional independence measure (FIM) [19,20]. In our study, we also used valid and reliable instruments to measure the risk of falls and functional capacities, in agreement with other studies that used TMT [21] and SPPB [16,20,27,29] in their assessments.

The exercise intervention included in the study consisted of adding two simple exercises (one for lower limb strength and other for balance) to the usual exercise routine carried out in a day care center. Day care centers are locations used to perform exercise programs for people with dementia [15,19,23] because they can spend a few hours a day doing activities aimed at maintaining their cognitive and physical abilities. Other studies’ interventions took place in nursing homes [18,23,27,28,29] or at home [14,16,17,20,26]. In our study, exercise group sessions with 10 people with dementia were guided by one physiotherapist for 45–50 min, 5 days a week. We intended to replicate the conditions in which exercise group sessions for people with dementia usually take place. We did not find any study where the ratio of patients per therapist was greater than the one in our intervention. Many similar studies have included home exercises and/or exercises guided and supervised individually by a therapist [14,15,16,17,18,19,20,21,22,23,25,26,27,28]. We consider it necessary to keep low ratios of people with dementia per physiotherapist in exercise group sessions because people with dementia frequently need help to follow group exercise sessions. This is in agreement with the study by Whitney et al. [23], with three patients per group, and with the study by Schwenk et al. [22], with four to six patients per group. However, keeping low ratios is not always possible in exercise group sessions in day care centers or nursing homes. We aimed to include an intervention that could be feasible in the facilities where people with dementia participate in non-pharmacological therapies such as exercise that can help maintain their abilities. 

Our study included a 5-week period of intervention. This may be a short period compared to other studies with interventions of 9 weeks [18,21], 10 weeks [24], 12 weeks [14,15,22,25,29], 16 weeks [28], 20 weeks [17,26], 24 weeks [23,27], or 1 year [16,19,20]. However, we aimed to analyze the effect of a simple and brief intervention on the risk of falls in the elderly with dementia. Therefore, five times a week is the frequency of exercise that we incorporated. We consider that in this way, we can help people with dementia move closer to the World Health Organization recommendations of physical exercise for people over 65 [30]. The number of sessions per week in similar studies we found is lower: 4 [18], 3 [27], 2 [15,21,22,23,24,25,29], or 1 [17,26]. Additionally, our intervention included exercise sessions of 45–50 min. This is similar to most of the studies we found, in which exercise sessions were between 30 and 60 min [15,16,17,18,19,20,21,23,26,27,28,29]. Only a couple of the studies we found included sessions of 2 h [22,25]. Finally, our intervention included strength and balance exercises, in accordance with most of the previous studies on exercise interventions on the risk of falls [14,16,18,19,22,23,25,27,28,29]. Some studies included special interventions such as therapy with dogs [15], Tai Chi classes [17,26], whole body vibration [21], or comprehensive motor–cognitive game-based training [24].

In our study, no significant difference was found between the groups in any of the assessments performed, although there was a tendency for the IG to score higher than the CG in the post-intervention assessments. This is in accordance with the studies by Taylor et al. [16], Whitney et al. [23], and Wesson et al. [14], where no significant difference between groups was found in the fall rate, balance score, or risk of falls after the intervention. Improvements in physical capacities and reductions in the risk of falls have been reported in other studies: walking endurance, leg muscle strength, balance [14,18], SPPB score [29], incidence of falls [25], and gait characteristics [22]. The main limitation of our study was the small sample size that was influenced by sick leave of potential participants, COVID-19 infection and restrictions, or lack of consent from caregivers to participate in the study. Nevertheless, our results showed a significant increase in scores for the Tinetti test and SPPB by comparing post- and pre-intervention assessments. Moreover, the higher the initial risk of falls, the higher the improvement obtained. It would have been interesting to test if the results could be maintained after a follow-up period.

## 5. Conclusions

Simple lower limb strength exercise and single leg stance feasible to be done in day care facilities are effective tools for reducing the risk of falls in older people with dementia, given that significant improvements were observed in the intervention group. This improvement was not influenced by sex, age, and years since the diagnosis of dementia.

## Figures and Tables

**Table 1 ijerph-19-11854-t001:** Sample characteristics.

Parameter	Group	Sex	Mean ± SD	*p* Value	Mean ± SD	*p* Value
Age (years)	IG	Men	71.0 ± 8.4	0.38	74.2 ± 8.9	0.70
		Women	76.3 ± 9.3			
	CG	Men	76.0 ± 4.5	0.92	75.6 ± 7.4	
		Women	75.4 ± 8.7			
BMI (Kg/m^2^)	IG	Men	28.8 ± 3.3	0.41	30.4 ± 4.6	0.55
		Women	31.4 ± 5.3			
	CG	Men	31.4 ± 5.1	0.27	29.2 ± 3.9	
		Women	28.2 ± 3.3			
Years since diagnosis	IG	Men	1.8 ± 1.7	0.42	2.5 ± 2.2	0.07
		Women	3.0 ± 2.5			
	CG	Men	5.0 ± 2.0	0.50	4.3 ± 2	
		Women	4.0 ± 2.1			
Barthel index score	IG	Men	91.3 ± 7.5	0.88	89.5 ± 8.0	0.79
		Women	88.3 ± 8.8			
	CG	Men	93.3 ± 2.9	0.78	91.0 ± 5.16	
		Women	90.0 ± 5.8			

Patients’ data and the comparisons between sexes (men vs. women) and groups (IG vs. CG). Student’s *t*-test for age, BMI, and years since diagnosis. Mann–Whitney U test for Barthel index score (*n* = 10 in IC, 6 women and 4 men; *n* = 10 in CG, 7 women and 3 men). SD = standard deviation, BMI = body mass index, IG = intervention group, CG = control group.

**Table 2 ijerph-19-11854-t002:** Scores for pre- and post-intervention assessments in IG and CG.

Test	Group	Pre-Intervention		Post-Intervention		Pre vs. Post
		Mean ± SD	*p* Value	Mean ± SD	*p* Value	*p* Value
TMT	IG	26.9 ± 0.6	0.64	27.6 ± 0.5	0.28	0.03
	CG	27.0 ± 1.1		27.1 ± 0.8		0.32
SPPB	IG	10.3 ± 1.0	0.87	11.8 ± 0.5	0.19	0.02
	CG	10.0 ± 2.1		11.3 ± 0.7		0.07

Functional ability scores and their comparisons between groups (IG vs. CG) and assessment (pre-intervention vs. post-intervention). Mann–Whitney U test for comparisons; Wilcoxon test for comparisons over time (*n* = 10 in IC, 6 women and 4 men; *n* = 10 in CG, 7 women and 3 men). TMT = Tinetti mobility test, SPPB = short physical performance battery, IG = intervention group, CG = control group.

**Table 3 ijerph-19-11854-t003:** Correlation between functional ability scores and patients’ characteristics.

Parameter	Group	TMT	SPPB
Sex	IG	χ^2^ = 2.2, *p* = 0.33	χ^2^ = 0.97, *p* = 0.91
	CG	χ^2^ = 2.86, *p* = 0.24	χ^2^ = 4.05, *p* = 0.54
Age	IG	r = 0.16, *p* = 0.67	r = −0.08, *p* = 0.83
	CG	r = 0.18, *p* = 0.62	r = 0.49, *p* = 0.15
Years since diagnosis	IG	r = 0.44, *p* = 0.20	r = 0.22, *p* = 0.53
	CG	r = 0.50, *p* = 0.14	r = 0.63, *p* = 0.05

Correlation indexes for post-intervention functional ability scores and the parameters sex, age, and years since diagnosis. Chi-squared (for sex) and Pearson’s correlation test (for age and years since diagnosis) (*n* = 10 in IC, 6 women and 4 men; *n* = 10 in CG, 7 women and 3 men). TMT = Tinetti mobility test, SPPB = short physical performance battery, IG = intervention group, CG = control group.

## Data Availability

Not applicable.

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
