# Peer review of "Feasible Intervention through Simple Exercise for Risk of Falls in Dementia Patients: A Pilot Study"

_ijerph, 2022, doi:10.3390/ijerph191911854_

Round 1
Reviewer 1 Report
This study by López García and Sánchez Ruiz focuses on the effectiveness of a simple exercise intervention in reducing the risk of falls in elderly with dementia. Although this is a preliminary study, with low number of patients and relatively short intervention period, authors find changes in TMT and SPPB when comparing pre- and post-intervention scores, suggesting that the exercise intervention assessed is a valid tool for reducing the risk of fall in these patients. The manuscript has been completed to a good standard.
Mayor concerns:
1) Despite dementia patients recruited were older than 60, their mean risk of falls was relatively low at the beginning of the intervention, based on the mean scores obtained (TMT > 25, SPPB > 10). In this regard, do the authors think that patients with higher initial risk of falling could generate more significant results?
2) In relation to the above, it would be interesting to analyze the possible correlation between the initial scores of the patients and the improvements obtained. Have the authors analyzed this possible relationship?
3) It would be good to know if there is a correlation between the scores obtained in the different tests in each of the assessments.
4) It would be interesting to know if there is a correlation between the improvements observed in the different tests used.
Minor concerns:
1) In the method section (line 94), the authors do not specify whether the physical therapist also guided the usual physical exercise program in the intervention group. Nor is it indicated who guided physical exercises in the control group.
2) In the method section (line 97), there is a mistake in the percentage values.
3) In the method section (line 98), it would be interesting to know to which group the patients with dementia other than Alzheimer´s disease belong.
Reviewer 2 Report
Introduction
1. It is better to put the description of the PA to the front, as the article focuses on the benefits of PA.
Materials and Methods
2. How was the sample size calculated?
3. As exercise is recommended in the initial phases of neurocognitive disorders (Silva et al., 2019), the dementia rating of the inclusion criteria was not specified, such as Clinical Dementia Rating (CDR 1 or 2), medical diagnoses. Moreover, a medical certificate attesting adequate fitness for the execution of exercise needed to be acquired.
References: Silva, F. O., Ferreira, J. V., Pl´acido, J., Sant’Anna, P., Araújo, J., Marinho, V., Laks, J., & Deslandes, A. C. (2019). Three months of multimodal training contributes to mobility and executive function in elderly individuals with mild cognitive impairment, but not in those with Alzheimer’s disease: A randomized controlled trial. Maturitas, 126, 28–33
4. What was the 5-week of the intervention duration determined based on?
5. A normality test needs to be used to determine the distribution of the data.
6. Randomization and blinding need to be showed.
7. The interactions between group (IG and CG) and time (baseline and follow-up) may be needed to test (Two-way repeated-measures ANCOVA), with sex, practice of PA as covariates.
Round 2
Reviewer 1 Report
Thank you for your response. No additional tasks are required.
Reviewer 2 Report
No comments and suggestions. Thanks.